# Early-Staged Carotid Artery Stenting Prior to Coronary Artery Bypass Grafting: Analysis of the Early and Mid-Term Results in Comparison with a Consecutive Cohort of Isolated Coronary Artery Surgery Patients

**DOI:** 10.3390/jcm13020480

**Published:** 2024-01-15

**Authors:** Paolo Nardi, Claudia Altieri, Calogera Pisano, Fabio Massimo Oddi, Alessandro Ranucci, Mauro Fresilli, Alessandro Cristian Salvati, Dario Buioni, Mattia Scognamiglio, Valentina Ajello, Carlo Bassano, Andrea Ascoli Marchetti, Arnaldo Ippoliti, Giovanni Ruvolo

**Affiliations:** 1Cardiac Surgery Division, Tor Vergata University Hospital, 00133 Rome, Italy; lindapisano82@gmail.com (C.P.); alessandrocristian.salvati@ptvonline.it (A.C.S.); docyuk@libero.it (D.B.); mattiasol@hotmail.it (M.S.); carlo.bassano@uniroma2.it (C.B.); giovanni.ruvolo@uniroma2.it (G.R.); 2Unit of Cardiology of the Cardiac Surgery Division, Tor Vergata University Hospital, 00133 Rome, Italy; claudia.altieri@ptvonline.it; 3Unit of Vascular Surgery, Tor Vergata University Hospital, 00133 Rome, Italy; fabio.massimo89@gmail.com (F.M.O.); ranuccialessandro01@gmail.com (A.R.); mauro.fresilli@gmail.com (M.F.); ascolimarchetti@med.uniroma2.it (A.A.M.); ippoliti@med.uniroma2.it (A.I.); 4Unit of Cardio-Thoracic Anesthesia, Tor Vergata University Hospital, 00133 Rome, Italy; valeajello@gmail.com

**Keywords:** carotid artery stenting, carotid endarterectomy, coronary artery bypass grafting, myocardial infarction

## Abstract

Aim: The aim of the present study was to analyze retrospectively the results of patients who underwent early-staged, i.e., within 24–48 h, carotid artery stenting (e-s CAS) before coronary artery bypass grafting (CABG). Methods: Between December 2014 and December 2022, 1046 consecutive patients underwent CABG; 31 of these patients (3%) were subjected to e-s CAS prior to CABG (e-s CAS + CABG group). Preoperative and intraoperative variables and early and mid-term results of the e-s CAS + CABG group were compared with those of patients who underwent isolated CABG (CABG group). Results: As compared with the CABG group, the e-s CAS + CABG group showed a worse clinical risk profile due to higher Euroscore-2 values and incidence of obstructive pulmonary disease and bilateral carotid artery and peripheral artery diseases (*p* < 0.05, for all comparisons). The combined end point of operative mortality, periprocedural myocardial infarction, and stroke was 3.2% (0%/0%/3.2%) in the e-s CAS + CABG group vs. 5.9% (2.2%/2.8%/0.9%) in the CABG group (*p* > 0.5, for all measurements). At 5 years, actuarial survival was 74% ± 16% in the e-s CAS + CABG group vs. 93% ± 4.0% in the CABG group, freedom from cardiac death was 100% vs. 98% ± 1.0% (*p* = 0.6), and freedom from MACCEs was 85% ± 15% vs. 97% ± 2.5% (*p* > 0.1, for all comparisons). Independent predictors of all-causes death were advanced age at the operation (*p* < 0.0001), a lower value for left ventricular ejection fraction (*p* = 0.05), and a high Euroscore-2 (*p* = 0.04). Conclusions: CABG preceded by e-s CAS appears to be associated with satisfactory early outcomes while limiting the risk of myocardial infarction to a very short time interval between the two procedures. Freedom from late all-causes death, cardiac death, and MACCEs were comparable and equally satisfactory, underscoring the positive protective effects of CAS and CABG on the carotid and coronary territories over time.

## 1. Introduction

Background. Almost 8% to 14% of patients who undergo coronary artery bypass grafting (CABG) are also affected by concomitant significant carotid artery stenosis [1]. Carotid artery disease, especially when a high-risk atherosclerotic plaque is present, represents an important risk factor for stroke after cardiac surgery, particularly after CABG. Stroke following CABG is mostly ischemic and related to embolic events, and ischemia injury is caused mainly by the subsequent hypo-perfusion that follows embolization. Embolization of cerebral arteries can be of particulate origin, i.e., derive from atherosclerotic and calcific material coming from a carotid plaque or from the wall of the ascending aorta or from the aortic arch, or be of gaseous origin, i.e., due to the entry of air bubbles from the circuits of the extracorporeal circulation. Ischemic stroke, if not of embolic origin, can result from cerebral hypo-perfusion during and after cardiac surgery [2,3,4,5]. Rationale of carotid revascularization. Concurrent severe carotid artery stenosis untreated during CABG has shown to confer a significant risk of perioperative stroke (>4%) and an increased risk of mortality and stroke in the mid-term follow-up (HR: 1.28–1.76) [6]. One of the main risk factors for cerebral hypo-perfusion following CABG is thought to be significant carotid disease, and therefore its treatment with either carotid endarterectomy (CEA) or carotid artery stenting (CAS) has the rationale to mitigate the risk of cerebral injury, both reducing the embolic risk and that of the hypo-perfusion due to the hemodynamic effect. Carotid endarterectomy can be performed either before or simultaneously (synchronously) with CABG [7]. Considering the high-risk clinical profile and the technical difficulties for the combined procedure of CEA + CABG, CAS can be a valid alternative for stroke prevention before CABG [8]. When performed in patients affected by significant carotid disease, CAS has shown a lower risk of myocardial infarction in comparison with CEA [9,10].

Moreover, CAS has been evolving in the last two decades as a valid alternative to traditional CEA [11,12], which undoubtedly remains the gold standard for the removal of an entire atherosclerotic plaque, as it has shown its effectiveness in high-risk patients and in the presence of concomitant coronary artery disease and heart failure, restenosis after CEA, and hostile neck anatomy. Numerous retrospective studies and meta-analyses have reported the results of the different carotid and coronary revascularization strategies, using synchronous or staged CEA + CABG, or with hybrid procedures using CAS + CABG, synchronously or staged. In particular, the end points analyzed were operative mortality and the incidence of stroke and myocardial infarction. However, despite the different strategies proposed, the optimal management of patients who undergo CABG in the presence of severe carotid artery disease still remains a matter of open debate. Objective. The aim of our study was to retrospectively analyze the cohort of patients affected by concomitant severe carotid and coronary artery ischemic disease who underwent CABG at our center in the last 8 years using the hybrid approach consisting of early-staged CAS (e-s CAS), i.e., performed within 24–48 h before CABG.

To better characterize the population object of our investigation, we compared the patients’ preoperative and perioperative clinical characteristics and results observed in the short- and mid-term periods with those of a consecutive cohort of patients who underwent isolated CABG in the same period of surgical activity.

We investigated if the early-staged approach thus understood, i.e., with a short interval of time from CABG, may better address the risk of neurological events within 24–48 h after carotid stenting and sought to clarify the benefits and risk ratios for bleeding complications, for kidney injury, and for periprocedural myocardial infarction.

## 2. Methods

From December 2014 to December 2022, at the Cardiac Surgery Division of the Tor Vergata University of Rome—Tor Vergata Polyclinic, 1046 patients underwent the intervention of CABG; 31 of these patients (3.0%) affected by severe carotid artery disease were subjected to carotid revascularization with early-staged carotid artery stenting (e-s CAS), 24–48 h prior to CABG. This patient population was the subject of our retrospective study.

Clinical features. To better characterize the clinical profile of patients suffering from concomitant coronary artery and carotid artery disease, we compared the clinical characteristics of this group (the e-s CAS + CABG group) with those of patients who underwent CABG alone (the CABG group). Clinical conditions were assessed at admission, i.e., the score risks of patients were calculated by the Euroscore-2 evaluation system for CABG (European System for Cardiac Operative Risk Evaluation) [13], and the indication to surgery was determined, i.e., elective, urgent, or emergency. In addition, the incidence of previous or recent, i.e., within 30 days, preoperative myocardial infarction, previous percutaneous coronary revascularization, and previous stroke, were evaluated. The cardiovascular risk factors analyzed were diabetes mellitus, especially insulin-dependent diabetes; smoking; dyslipidemia; and hypertension.

Associated pathologies. The following pathologies were considered: chronic obstructive pulmonary disease, expressed with a forced expiratory volume in 1 s < 75% of the normal value or requiring bronchodilators; peripheral arterial disease, considered to be significant in the presence of claudication with autonomous walking < 200 m and hyposphygmia pulses; carotid artery disease, considered to be significant in the presence of stenosis > 50% detected by ultrasound echo-color Doppler; severe carotid artery, disease considered to be significant in the presence of stenosis > 70%; and chronic renal dysfunction, defined as moderate in the presence of creatinine clearance 50–80 mL/min. or severe with creatinine clearance < 50 mL/min.

Anatomical and hemodynamic data. The extent of coronary artery pathology was assessed on the basis of coronary angiography. Left ventricular function, expressed as left ventricular ejection fraction, was evaluated by preoperative transthoracic and intraoperative trans-esophageal echocardiography in all patients.

### CAS Procedure and Anti-Platelet Therapy Administration

All patients affected by coronary artery disease were subjected routinely to an echocardiographic color-Doppler duplex ultrasonography scanner and a Carotid Duplex Ultrasound (DUS). In our Institute, until 2008 [14], CAS prior to CABG was indicated for an internal carotid artery stenosis greater than 50% in symptomatic patients or at least 80% in asymptomatic ones, according to the North American Symptomatic Carotid Endarterectomy Trial [15] and the Committee for the Asymptomatic Carotid Atherosclerosis Study [16]. Subsequently, we extended the hybrid treatment to patients with an internal carotid artery stenosis greater than 70%, regardless of symptoms. Patients with the indication for carotid artery stenting also underwent CT angiography of the epi-aortic vessels and head CT.

Aspirin, 100 mg daily, was started at least 2 days before CAS, and clopidogrel was administered as a loading dose, 300 mg, the day before CAS and continued together with aspirin at a dosage of 75 mg after CAS. After 1–2 days, CABG operations were performed with patients on dual antiplatelet therapy. In the last 3 years, the loading dose of clopidogrel to be taken before CAS has been reduced from 300 mg to 150 mg.

Heparin was administered at a dose of 5000 I.U. intravenously to prevent thrombosis before the carotid stenting procedure. CAS was performed through percutaneous trans-femoral access under local anesthesia with the use of cerebral protection devices in all cases. At the end of the CAS procedure, the patients were subsequently transferred to the in-patient department and monitored for 24–48 h after by continuous ECG and repeated measurements of blood pressure and heart rate, taking dual antiplatelet therapy with aspirin 100 mg plus clopidogrel 75 mg daily.

At the end of CABG, following heparin antagonization with protamine hydrochloride, aspirin 100 mg plus clopidogrel 75 mg was re-started in the intensive care unit, and its administration was performed by naso-gastric tube or after extubation, providing that blood drainage from the thoracic tubes was less than 50 mL/h.

Clopidogrel was then continued orally, 75 mg daily, for one month following CAS.

Surgical strategy. A complete sternotomy was always performed. Coronary artery bypass grafting was performed by means of cardiopulmonary bypass and ascending aorta cross-clamping. Cardiac arrest was achieved using intermittent antegrade warm blood cardioplegia, 600 mL as the first dose, followed by 400 mL doses administered every 20–25 min, or with the use of St. Thomas cold crystalloid solution, 10 mL/kg as the first dose, followed by 5 mL/kg doses administered every 30–35 min, as previously reported [17].

The left internal mammary artery was used as a graft for the left anterior descending artery in association with saphenous vein grafts for the revascularization of the right coronary artery and/or to the left circumflex artery branches. The use of the autologous saphenous vein alone without internal mammary artery harvesting was reserved for patients suffering from severe pulmonary disease, i.e., in the presence of diffuse bronchiectasis and emphysematous pathology, in the presence of proximal stenosis of the subclavian artery, or when serious hemodynamic and or electrical instability was present at the anesthetic induction.

Definitions and data analysis. The current study considered all consecutive patients who underwent CABG as an isolated procedure with or without subsequent e-s CAS. Other CABG procedures performed in association with heart valve repair or replacement, or in association with ascending aorta repair, were not included.

The study was approved by the Institutional Review Board of the Tor Vergata Polyclinic (protocol code no.: 108/23, date of approval: May 2023). All patients gave their informed surgical consent. The study was retrospective.

Operative or in-hospital mortality included deaths occurring in the hospital or within 30 days after CAS + CABG or isolated CABG operations. The main cardiac and non-cardiac postoperative complications analyzed were perioperative myocardial infarction, defined as an increase in serum troponin I > 20 ng/mL associated with an increase in the creatine-kinase muscle-brain CK-MB enzyme greater than 10% of the total CK enzyme and the onset of ECG anomalies from acute ischemia. Low cardiac output syndrome was defined in the presence of a cardiac index <2.0 L/min/m^2^ with or without renal impairment requiring inotropic support for a period greater than 48 h.

A neurological complication was defined as an episode of stroke due to a focal or general cerebral lesion; acute renal impairment was defined as a two-fold increase in preoperative serum creatinine level or oliguria necessitating continuous veno-venous filtration.

Respiratory failure was defined as an episode of primary respiratory insufficiency requiring mechanical support for a period greater than 48 h, tracheal re-intubation, or intermittent application of non-invasive positive-pressure ventilation.

Follow-up evaluation was performed with an outpatient visit to the patient and/or by telephone interview at 34 ± 25 (median: 31, range: 3–84) months. The following adverse cardiac and cardio-cerebrovascular events were evaluated over time: death for all causes; cardiac death; and major adverse cardiac and cerebrovascular events (MACCEs) rate, defined as death from cardiac causes, myocardial infarction onset with or without the need for new coronary artery revascularization, and stroke onset. The follow-up ended in August-September 2023.

Statistical analysis was performed with the use of Stat View 4.5 (SAS Institute Inc., Abacus Concepts, Berkeley, CA, USA). Contingency table raw data were calculated with the use of chi- and G-squared and Fisher’s exact tests for categorical variables and the unpaired Student’s *t*-test for continuous variables to perform the comparisons of early-staged CAS plus CABG and CABG patients. The analyzed variables were age; gender; Euroscore-2 expressed as a mean value and divided into patients at low risk (ES 1–3%, L), at medium risk (ES > 3 ≤ 7%, M), and at higher risk (ES > 7%, H); clinical presentation, i.e., previous and recent myocardial infarction; stable or unstable angina; left ventricular ejection fraction; body mass index; height; and weight. Preoperative variables also included comorbidity, i.e., smoking, diabetes mellitus, insulin-dependent diabetes mellitus, hyperlipidemia, hypertension, chronic obstructive pulmonary disease, chronic renal dysfunction, carotid and peripheral arterial disease, and indication to CABG, i.e., elective, urgent, or emergency. Intraoperative analyzed variables included the number of grafts per patient, cardiopulmonary bypass and aortic cross-clamp times, and the use of left internal mammary artery grafting. Five survival and event-free survival curves were calculated. Late survival, not including in-hospital mortality, and freedom from late cardiac death and from MACCEs were calculated by the means of the Kaplan–Meier method and were expressed as mean values of percentages ± 1 standard deviation. The Mantel–Cox log-rank and Breslow–Gehan–Wilcoxon rank tests were used to compare the curves of freedom from events, i.e., among CAS + CABG vs. CABG patients and among patients with and without a specific Euroscore-2 risk profile. Cox proportional regression analysis was used to evaluate the influence of the variables on time to death. All continuous values were expressed as means ± 1 standard deviation. All *p*-values < 0.05 were considered statistically significant.

## 3. Results

Clinical presentation and associated co-morbid conditions. As compared with the CABG group, the e-s CAS plus CABG group at admission had higher Euroscore-2 values and higher incidence of obstructive pulmonary disease, bilateral carotid artery disease, and peripheral arterial disease (*p* < 0.05, for all comparisons) (Table 1).

Intraoperative data and perioperative results. Eighty-five CABG operations (8.1%) were off-pump. The rate of use of left internal mammary artery grafting was 87% for the e-s CAS + CABG group and 95% for the CABG group (*p* < 0.05) (Table 2). The lower rate of internal mammary artery use observed in the CAS + CABG group was primarily driven by the presence of the higher incidence of stenosis of the proximal segment of the subclavian artery.

Operative mortality was 2.1% in the whole population (22/1046): it was absent in the e-s CAS + CABG group and 2.1% (22/1015) in the CABG group (*p* = 0.5).

The combined end point of in-hospital death/perioperative myocardial infarction/stroke was 3.2% in the e-s CAS + CABG group (0%/0%/3.2%) vs. 5.9% in the CABG group (2.2%/2.8%/0.9%) (*p* > 0.5, for all measurements). In the e-s CAS + CABG group, only one patient suffered a stroke upon awakening after CABG surgery, while he had not presented any neurological symptoms during CAS and in the subsequent hours before undergoing CABG. The brain lesion was ipsilateral to the untreated contralateral carotid artery, chronically occluded.

Low cardiac output syndrome onset and acute worsening renal function were absent in the e-s CAS + CABG group and 3.8% (39/1015) and 4.0% (41/1015) in the CABG group, respectively (*p* > 0.5, for both comparisons). Surgical re-exploration for bleeding was absent in the e-s CAS + CABG patients and 2.0% in the other group of patients (*p* > 0.6) (Table 2, second part).

Late Results. Follow-up was 98% complete; 19 patients were lost in the CABG group. There were 47 deaths out of 986 patients who completed the follow-up. At 5 years, actuarial late survival was 74% ± 16% for the e-s CAS + CABG patients vs. 93% ± 4.0% in the CABG patients (*p* = 0.12), freedom from cardiac death was 100% vs. 98% ± 1.0% (*p* = 0.56) (Figure 1), and freedom from MACCEs was 85% ± 15% vs. 97% ± 2.5% (*p* = 0.38) (Figure 2).

The concomitant carotid stenting procedure was not detected as a risk factor for cardiac death or reduced freedom from adverse cardiac and cerebrovascular events in both Cox linear and multivariate Cox regression analyses. In the Cox regression analysis, independent predictors of all-cause late death were advanced age at the operation (*p* < 0.0001), a lower preoperative left ventricular ejection fraction (*p* = 0.05), and a higher value (H) for Euroscore-2 (*p* = 0.04) (Table 3).

Advanced age at the operation remained the only independent predictor of reduced freedom from cardiac death (*p* = 0.05) (Table 4). At five years, the rank-test analyses showed significantly reduced survival in patients with high-risk Euroscore-2 (high: 59% ± 13% vs. medium: 81% ± 5.4% and low: 96% ± 1.0%, *p* < 0.0001) (Figure 3). Functional classifications improved from preoperative to follow-up values: CCS class improved from 2.8 ± 1.0 to 1.0 ± 0.4 (*p* < 0.0001) and NYHA class improved from 1.8 ± 1.0 to 1.5 ± 0.5 (*p* < 0.01). Internal carotid artery blood flow, assessed by duplex-scanner echocardiographic color-Doppler ultrasound, showed a very significant improvement in the patients who underwent carotid artery stenting in comparison with the preoperative ones (60 ± 10 cm/s vs. 290 ± 30 cm/s; *p* < 0.0001).

## 4. Discussion

It is well known that the presence of severe carotid artery disease associated with ischemic heart disease makes the patient undergoing CABG at greater risk of adverse perioperative cardiovascular and cerebrovascular events, in particular, death, myocardial infarction, and stroke. In fact, the preoperative clinical presentation of patients who underwent CAS associated with CABG that we observed was worse in comparison with patients who underwent isolated CABG: higher Euroscore-2, i.e., approximately 1% increased risk, and a markedly higher incidence of peripheral arterial diseases and chronic pulmonary disease (Table 1). While, on the one hand, carotid revascularization undoubtedly has the aim of reducing or minimizing the risk of postoperative stroke, on the other hand, this procedure “per se” increases the risk of adverse cardiac events during the entire perioperative period and postoperatively.

Different surgical strategies, i.e., CEA and CABG, synchronously or staged, and, subsequently, interventional strategies, i.e., CAS + CABG (one-day or same-day) and staged CAS prior to CABG, have been proposed and examined with large meta-analyses to try and establish which of these may be the best option for these patients.

Haywood and coworkers, for a population of patients who underwent synchronous CEA + CABG (*n* = 634) in comparison to staged CEA − CABG (*n* = 309), reported a similar incidence of perioperative stroke (4.6% vs. 4.1%) and death (7.0% vs. 5.0%) but a lower rate of MI (5.5% vs. 11.5%, *p* < 0.01) [18]. Modugno et al., in 222 combined CEA + CABG procedures, reported 4.1% mortality and stroke [19]. Sharma et al., in a meta-analysis involving 17,469 synchronous CEA + CABG patients and 7552 staged CEA − CABG patients, showed no difference in early mortality (4% vs. 3.4%), stroke (4.3% vs. 1.9%), combined end-point mortality/stroke (7.9% vs. 6.2%), and MI/stroke (6.9% vs. 10.1%) [20]. As can be seen from the data reported above and published recently, it appears clearly that the operative mortality in combined surgical operations, both synchronous and staged, is approximately double, if not higher, than in isolated CABG operations, as well as the risk of stroke and myocardial infarction. For example, in our series, the incidence of infarction and stroke in isolated CABG was below 3% and 1%, respectively. Patients who undergo the synchronous surgical procedure likely face additional cardiovascular stress related to the prolonged exposure to anesthesia, the clamping of the carotid artery during the endarterectomy, and the extracorporeal circulation, thus compromising the adequacy of cerebral perfusion. On the other hand, a staged surgical procedure can mitigate these risks, but the time interval between CEA and CABG can expose the patient to an incremental risk of MI due to the severity of the ischemic disease. For these reasons, in the last 20 years, carotid stenting has found widespread use to treat patients undergoing CABG in order to eliminate the surgical stress typical of CEA in patients also suffering from severe coronary artery disease.

The Patients at High Risk for Endarterectomy (SAPPHIRE) trial showed that CAS was safer than CEA and that it lowered the risk of MI within 30 days after the procedure [21]. Since then, several studies have focused on the comparison between staged CEA-CABG, traditional synchronous CEA + CABG, and deferred CABG, i.e., CAS prior to CABG.

Giannopoulos and coworkers [22], in an extensive meta-analysis of five studies including 16,712 patients subjected to synchronous CEA + CABG (*n* = 15,727) and staged CAS − CABG (*n* = 985), showed no differences in terms of stroke (3% vs. 3%) and MI (5% vs. 5%), while the risk of mortality was higher in patients who underwent CEA plus CABG synchronously (4% vs. 2%). Feldman et al. [23], during a 9-year period of study in which 15,402 (68.4%) synchronous CEA + CABG procedures, 6297 (28%) staged CEA − CABG procedures, and 802 staged CAS − CABG procedures were performed, observed an increased risk of death for both surgical carotid revascularizations in comparison with CAS (HRs: 2.08 and 2.40, respectively) but a lower risk of stroke. Interestingly, the risk of stroke and death for the CAS procedures performed in the last 4-year period decreased significantly. Shishehbor and coworkers [24], in a population of 350 patients who underwent carotid revascularization within 90 days before cardiac surgery, in the early phase did not show any differences in the risk of death, stroke, and MI for CEA + cardiac surgery and staged CAS—cardiac surgery, but, on the contrary, staged CEA—cardiac surgery incurred a higher rate of MI. Subsequently, staged CAS + cardiac surgery patients experienced fewer adverse events compared with both synchronous and staged CEA patients.

From these reported studies, it can be seen that CAS, in patients suffering from concomitant ischemic carotid and coronary disease, is equally effective in comparison with CEA in the prevention of perioperative stroke and that it reduces the risk of operative mortality more significantly, making it similar to that observed for patients undergoing isolated CABG.

For these reasons, over the years, we have also modified the approach to combined carotid and coronary artery revascularization, preferring the CAS procedure over traditional CEA. From 2005 to 2013, we initially proposed and performed the same-day synchronous CAS + CABG protocol. In a study population of 97 patients who underwent same-day CAS + CABG, we observed a 2.1% rate of in-hospital death, a 1% rate of transient ischemic attack occurrent during the CAS procedure, and no episodes of MI [14]. Using this protocol, dual antiplatelet therapy was started at the end of CABG, while the CAS procedure was carried out only in single antiplatelet therapy with ongoing aspirin, 100 mg daily. Our findings are in accordance with those reported by Zivkovic et al. [25]: in 69 patients who underwent same-day CAS + CABG, the authors reported 1.4% MI, 1.4% stroke, and no mortality. The main limitation of the mentioned protocol can be represented by the use, within a single day, of contrast dye to perform CAS and, subsequently, extracorporeal circulation, both of which increase the risk of acute renal impairment. We observed this complication in six (4.5%) cases. Considering the fact that approximately 30–40% of patients who underwent CABG in our series were affected by moderate or severe chronic renal dysfunction, the risk of experiencing acute renal failure was not at all negligible, and it can determine a substantial risk of death in the short and in the medium term. Another limitation of this protocol may reside in the fact that dual antiplatelet therapy was started only after CABG surgery, thus leaving the time interval between the carotid stenting procedure and the CABG uncovered by dual antiplatelet therapy, potentially increasing the risk of stroke in that interval of time. Then, since 2014, we have adopted the staged approach based on CAS only 1–2 days prior to CABG surgery, in any case reserving a very narrow interval between the two procedures (i.e., early staging). That choice was based on the rationale to minimize or virtually eliminate the risk of MI typically instead of in all staged procedures in which CABG is performed several days or even several weeks [22,24,26] after CAS or CEA.

With this strategy, it is also possible to observe whether neurological events occur in the 24–48 h after the CAS procedure before CABG is performed and whether the risk of renal damage linked to the double insult on the same day of the contrast medium and the cardiopulmonary bypass is likely to be reduced. Although the number of patients is limited for now, we have not observed any complications of this type (Table 2).

Regarding the risk of bleeding, from a meta-analysis of 2727 patients subjected to same-day CAS + CABG or CAS and then CABG, a significant difference in the incidence of 30-day death/stroke was not highlighted (5.9% vs. 8.5%). However, it was highlighted that the risk of postoperative bleeding was greater in the group of patients with single antiplatelet therapy before CAS and then dual therapy after CABG compared to those in dual therapy before CAS reduced then to single therapy before CABG (7.3% vs. 2.8%) [27]. In our series, we did not observe a higher risk of bleeding in such patients compared to the population of patients who underwent isolated CABG (0% vs. 2.0%). Furthermore, in the last 5 years, the need to urgently perform CABG on patients undergoing single or double anti-aggregating therapy has become increasingly frequent, due to the high-risk coronary anatomy and because in clinical practice many patients have intracoronary stents positioned shortly before carrying out CABG, thus necessarily requiring the dual antiplatelet therapy. Anyway, we observed in a previously published study [28] that bleeding requiring surgical exploration in patients on dual antiplatelet therapy (aspirin plus clopidogrel), i.e., 44% of the sample, although higher (2.4% vs. 1.2%), did not increase the risk of in-hospital mortality.

At the mid-term follow-up, we observed a high rate of event-free survival (Figure 1 and Figure 2) and satisfactory late survival and functional classifications in all patients. Five-year survival was negatively affected by worse preoperative clinical presentation (i.e., advanced age, a higher Euro-score, and a lower left ventricular ejection fraction) and not by the type of procedure performed (Table 3 and Table 4) (Figure 3).

Brott et al. [11], in 2502 patients enrolled in the CREST trial, at 10 years reported a limited and similar incidence of stroke after CEA and CAS, i.e., 5.6% and 6.9%, respectively.

During follow-up, we observed 2 episodes of strokes, 15 cardiac ischemic events requiring or not requiring coronary stent implantation, 14 cardiac-related deaths in the isolated CABG patients, 1 non-cardiac-related death, and 1 cardiac ischemic event requiring coronary stent implantation in patients in the e-s CAS + CABG group, thus showing satisfactory and similar freedom from MACCEs. As expected, the beneficial effect of myocardial revascularization on freedom from cardiac death was evident and equally satisfactory in both groups (Figure 1). These findings are substantially similar to those observed in other studies in short-term [11,25] and in mid-term follow-up periods [29].

### Strengths and Limitations of the Study

The strengths of our study are represented by the fact that CABG preceded by e-s CAS is associated with very limited operating risk and perioperative complications, comparable to those observed for patients operated on with isolated CABG, while at the same time limiting the risk of MI in a narrow time interval of 24–48 h between the two procedures.

Some limitations are present in our study. It was a retrospective, non-randomized, non-propensity score-matched study. Moreover, the study included patients from a single center only, although in the mentioned period of surgical activity the surgeons who treated the patients remained the same, and therefore there was no bias regarding the surgical approaches and techniques used. The major limitation is that the e-s CAS + ABG group represented only 3% of all surgeries, limiting the power of our study. However, to better characterize the patient population, we carried out a comparison of preoperative and intraoperative variables, analyzing in depth the main, early, and late cardiovascular outcomes and comparing them with those of more recently published studies which have addressed the issue in question.

## 5. Conclusions

From what was observed in our analysis, and in light of what is still reported today by a rather well-represented literature, all carotid and coronary artery revascularization surgical and interventional hybrid strategies can be used with relative safety and effectiveness, with each having specific advantages with respect to specific clinical patient conditions, in order to obtain the best possible results.

The staged surgical procedures, with a too-long time-interval between CEA and coronary artery bypass surgery, are associated with a substantially high MI risk.

Generally, carotid stenting, both synchronous (same day) and staged, as the procedure of choice in such complex patients, is preferred because it has been shown to reduce cardiovascular adverse events, especially operative mortality, at the expense of a greater potential risk of postoperative bleeding. On the contrary, e-s CAS allows for better monitoring and the possible treatment of any neurological and renal procedural complications, while limiting the risk of myocardial infarction to a very narrow time interval. To confirm this, it will be necessary to expand the study.

Freedom from late all-causes mortality, cardiac death, and adverse cardiac and cerebrovascular events are comparable and equally satisfactory, underscoring the positive protective effect of CAS and CABG on the carotid and coronary territories over time.

## Figures and Tables

**Figure 1 jcm-13-00480-f001:**
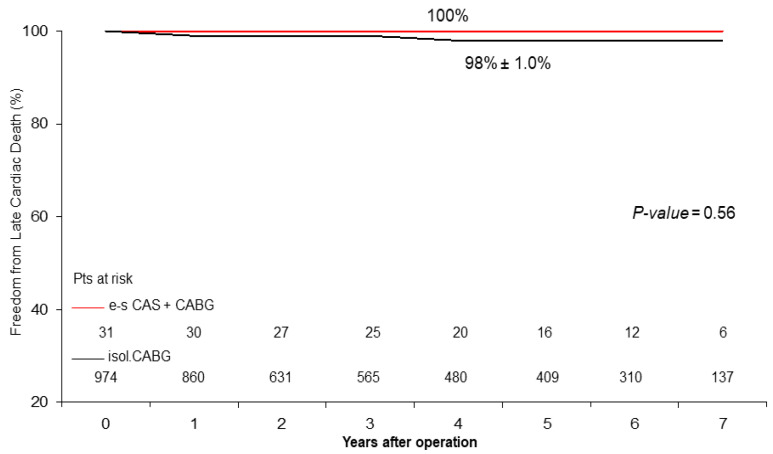
Freedom from late cardiac death after early-staged CAS + CABG operation and isolated CABG (mean follow-up: 34 ± 25 (M 31) months).

**Figure 2 jcm-13-00480-f002:**
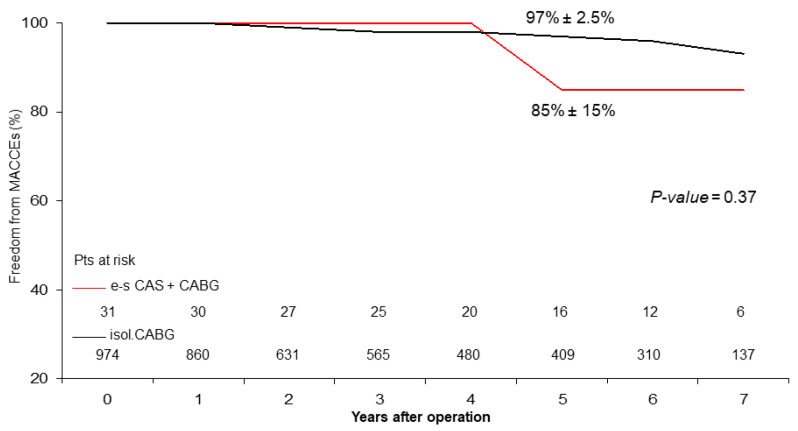
Freedom from MACCEs (major adverse cardiac and cerebrovascular events).

**Figure 3 jcm-13-00480-f003:**
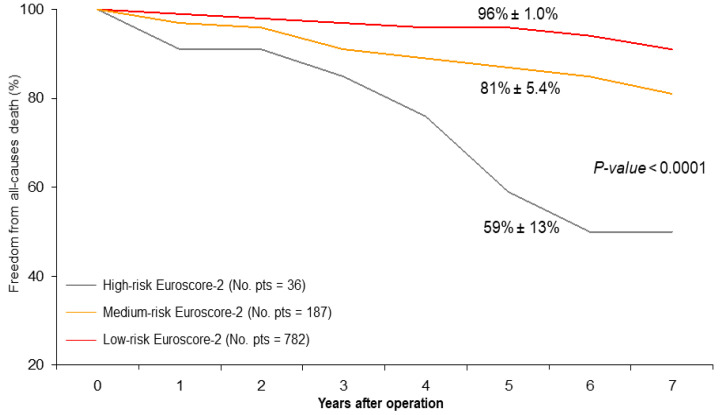
Mid-term survival stratified by the Euroscore-2 risk profiles (low, medium, or high) (log-rank Mantel–Cox and Breslow–Gehan–Wilcoxon tests, all *p*-values < 0.0001).

**Table 1 jcm-13-00480-t001:** Clinical preoperative characteristics and associated pathologies in patients who underwent early-staged CAS plus CABG (e-s CAS + CABG group) in comparison with patients who underwent isolated CABG (CABG group).

Characteristics	e-s CAS + CABG Group (*n* = 31)	CABG Group (*n* = 1015)	*p*-Value
Age, years	68.2 ± 6.6	67.3 ± 9.0	0.668
EuroSCORE-2, %	3.20 ± 0.14	2.53 ± 0.17	0.05
Previous MI (>30 days), *n* (%)	32 (20)	257 (25.3)	0.912
Recent MI (<30 days), *n* (%)	6 (19.3)	203 (20.0)	0.925
Left ventricular ejection fraction, %	50.6 ± 8.9	52.3 ± 8.8	0.531
Congestive heart failure history, *n* (%)	10 (32.2)	344 (33.9)	0.630
Previous stroke, *n* (%)	2 (6.45)	55 (5.42)	0.712
Previous percutaneous coronary stent/s, *n* (%)	4 (12.9)	125 (12.3)	0.960
Indication to CABG, *n*(%):			0.979
Elective	7 (23)	247 (23)	
Urgent	24 (77)	750 (74)	
Emergency	- (0)	18 (1.8)	
Diabetes mellitus, *n* (%)	10 (32.3)	306 (30.1)	0.261
Diabetes mellitus (insulin-dependent), *n* (%)	9 (29.0)	295 (29.3)	0.702
Hypertension, *n* (%)	28 (90.0)	883 (87.0)	0.343
Hypercholesterolemia, *n* (%)	22 (70.9)	690 (67.9)	0.717
Smoking, *n* (%)	23 (74.2)	721 (71.0)	0.293
Weight, kg	75.0 ± 7.7	77.7 ± 13.0	0.612
Height, m	1.03 ± 0.7	1.17 ± 0.7	0.493
Body mass index	26.1 ± 2.2	27.1 ± 4.1	0.402
Chronic pulmonary disease, *n* (%)	8 (25.8)	81 (7.98)	0.01
Severe renal dysfunction, *n* (%)	1 (3.23)	91 (9.06)	0.625
Moderate renal dysfunction, *n* (%)	12 (38.7)	344 (34.0)	0.429
Bilateral carotid artery disease, *n* (%)	13 (41.9)	71 (6.99)	<0.0001
Peripheral arterial disease, *n* (%)	15 (50)	72 (7.09)	<0.0001

CAS: carotid artery stenting; CABG: coronary artery bypass grafting; MI: myocardial infarction.

**Table 2 jcm-13-00480-t002:** Perioperative variables and perioperative outcomes.

Variables	e-s CAS + CABG Group (*n* = 31)	CABG Group (*n* = 1015)	*p*-Value
Cardiopulmonary bypass, minutes	90.9 ± 33.3	99.1 ± 61.8	0.663
Aortic cross-clamp, minutes	56.6 ± 23.1	58.1 ± 32.8	0.885
No. of grafts per patient, mean v.	2.8 ± 0.5	2.8 ± 0.8	0.779
L-ITA use, *n* (%)	27 (87)	837 (95)	0.02
Completeness of revascularization, *n* (%)	30 (97)	995 (98)	0.877
Combined end point (in-hospital death/MI/stroke), *n* (%)	1 (3.20)	59 (5.80)	0.769
In-hospital mortality, *n* (%)	(0)	22 (2.2)	
MI, *n* (%)	(0)	28 (2.8)	
Stroke, n (%)	1 (3.20)	9 (0.9)	
Acute heart failure, *n* (%)	(0)	39 (3.8)	0.431
Acute renal dysfunction, *n* (%)	(0)	41 (4.0)	0.418
Respiratory failure, *n* (%)	2 (6.45)	147 (11.2)	0.432
Surgical re-exploration for hemostasis, *n* (%)	(0)	21 (2.0)	0.640

L-ITA: left internal thoracic artery; MI: myocardial infarction.

**Table 3 jcm-13-00480-t003:** Risk factors and independent predictors of late survival.

Variables	*p*-Value (Cox Linear)	Hazard Ratio	95% CI	*p*-Value (Cox Regression)
Advanced age at the operation (74.5 vs. 66.9 years)	<0.0001	4.12	1.05–1.16	<0.0001
Reduced LVEF (48.5% vs. 52.6%)	0.0027	−1.95	0.93–1.00	0.050
Euroscore-2 (H group vs. M and L groups)	0.001	2.09	0.11–0.93	0.036

LVEF: left ventricular ejection fraction.

**Table 4 jcm-13-00480-t004:** Risk factors and independent predictors of late cardiac death.

Variables	*p*-Value (Cox Linear)	Hazard Ratio	95% CI	*p*-Value (Cox Regression)
Advanced age at the operation (72.9 vs. 67.2 years)	<0.0001	1.92	0.99–1.51	0.05
Reduced LVEF	0.0928	−1.25	0.91–1.02	0.212
EuroSCORE (H risk group vs. M and L risk groups)	0.0628	1.21	0.12–12.1	0.867

LVEF: left ventricular ejection fraction.

## Data Availability

Surgical and clinical data were obtained from the database of the Tor Vergata Polyclinic.

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
