# Peer review of "Early-Staged Carotid Artery Stenting Prior to Coronary Artery Bypass Grafting: Analysis of the Early and Mid-Term Results in Comparison with a Consecutive Cohort of Isolated Coronary Artery Surgery Patients"

_jcm, 2024, doi:10.3390/jcm13020480_

Round 1
Reviewer 1 Report
Comments and Suggestions for Authors
This study analyzes the outcomes of patients who underwent carotid artery stenting (CAS) 24–48 hours before coronary artery bypass grafting (CABG) (early, es). Conclusions: CABG preceded by early CAS surgery appears to be associated with satisfactory early outcomes while limiting the risk of myocardial infarction to a very short time interval between the two surgeries. Avoidance of late all-cause death, cardiac death, and MACCE were comparable and equally satisfactory, underscoring the positive protective effects of CAS and CABG on the carotid and coronary territories over time.
Overall, the novelty, concept, and manuscript writing of this study are of a certain publication standard.
Author Response
- To Reviewer 1: we thank very much the Reviewer 1 for his positive evaluation of the manuscript. In the text, we have added in red colour further aspects and details in the Methods section for the carotid stenting procedure (lines 119-120; and line 133).
The Figure 2 has been correctly inserted and explained in the Results section (line 253).
The Discussion section, at its end, was changed in accordance with the comments of the Reviewer 2 (see also in red colour the changes, line 404 – continues).
The conclusions of the abstract (line 34, onwards) and of the manuscript (line 414, onwards) have been modified in according with your suggestions.
Thank you very much again.
Reviewer 2 Report
Comments and Suggestions for Authors
The study titled "Early-staged carotid artery stenting prior to coronary artery bypass grafting: Analysis of the early and mid-term results in comparison with a consecutive cohort of isolated coronary artery surgery" conducted by Nardi et al. compared the outcome of CABG with CABG+CAS. The study is well-designed and provides helpful information on the topic. I have some major concerns:
1- The study is single-center. As the conduction period is more than 5 years (2014-2022), many changes like surgeons in your center may impact your results. You should clearly mention this limitation in the discussion and suggest future studies to overcome this important limitation.
2- I found several errors in using abbreviated forms. You MUST define all abbreviations in their first use and re-check that you are using ONLY abbreviated forms after the definition of each abbreviation.
3- The CAS+CABG group is only 3% of all your surgeries, limiting the power of your study. I suggest adding a propensity score matching analysis, if possible. Otherwise, clearly add this limitation to the limitations section.
4- Add the strengths of your study prior to the limitations section. I suggest using the "Strengths and limitations" heading separate from the discussion.
5- Please check the manuscript for typos and grammatical errors.
Comments on the Quality of English LanguageI found several errors in using abbreviated forms. You MUST define all abbreviations in their first use and re-check that you are using ONLY abbreviated forms after the definition of each abbreviation.
Please check the manuscript for typos and grammatical errors.
Author Response
- we thank very much the Reviewer 2 for his suggestions and comments to implement the manuscript. 1) in the mentioned period of surgical activity reported in the manuscript the surgeons who treated patients remained the same, and therefore there was no bias regarding the surgical techniques used in the entire period. We have included and explained this aspect in the new chapter at the end of the discussion (separated) entitled as you suggested: "Strengths and Limitations" (see in red colour).
Second point: The abbreviations have been better defined in the abstract sections and in the Text, and always used in the manuscript (e-s CAS; e-s CAS + CABG group; CABG group) and in Table 1 and 2 (line 218; line 228).
Third point: We have explained these limitation in the new chapter at the end of the discussion entitled as you suggested: "Strengths and Limitations" (from line 404- onwards, see in red colour).
Fourth point: we have added a new chapter to better explain the strengths of the study and its limitations (from line 404-onwards).
Five point: The Figure 2 has been correctly inserted and explained in the Results section (line 253), and errors in English language have been checked.
Thank you very much again for your revision.
Round 2
Reviewer 2 Report
Comments and Suggestions for Authors
Thanks for revisions.